# Comparative Mutagenic Effectiveness and Efficiency of Gamma Rays and Sodium Azide in Inducing Chlorophyll and Morphological Mutants of Cowpea

**DOI:** 10.3390/plants11101322

**Published:** 2022-05-16

**Authors:** Aamir Raina, Rafiul Amin Laskar, Mohammad Rafiq Wani, Basit Latief Jan, Sajad Ali, Samiullah Khan

**Affiliations:** 1Mutation Breeding Laboratory, Department of Botany, Aligarh Muslim University, Aligarh 202002, India; khan_drsami@yahoo.co.in; 2Botany Section, Women’s College, Aligarh Muslim University, Aligarh 202002, India; 3Department of Botany, Bahona College, Jorhat 785101, India; rafihkd@gmail.com; 4Department of Botany, Abdul Ahad Azad Memorial Degree College Bemina, Cluster University Srinagar, Jammu and Kashmir 190018, India; botanyrafiq@gmail.com; 5Department of Clinical Pharmacy, College of Pharmacy, King Saud University, Riyadh 11451, Saudi Arabia; basitlatief@gmail.com; 6Department of Biotechnology, Yeungnam University, Gyeongsan 38541, Korea; sajadmicro@yu.ac.kr

**Keywords:** *Vigna unguiculata* (L.), mutation frequency, seed germination, pollen sterility, coefficient of interaction, M_1_/M_2_ generations

## Abstract

Mutagenic effectiveness and efficiency are the most important factors determining the success of mutation breeding, a coherent tool for quickly enhancing genetic diversity in crops. However, conclusive evidence of using an effective and efficient dose of gamma (γ) rays and sodium azide (SA) for genetic improvement is scant. The present study assesses genetic diversity in M_2_ mutants of cowpea and evaluates mutagenic effectiveness and efficiency of the single and combination doses of γ rays and SA. In M_0_ generation, 7200 M_1_ seeds obtained by SA treatment (0.01—0.1%) and γ irradiation (100—1000 Gy) at a dose rate of 11.58 Gy/min were sown to raise M_1_ generation. A total of 57,620 M_2_ seeds were generated from the M_1_ generation of two varieties—Gomati VU-89 and Pusa-578, from which 47,650 seeds germinated. Moreover, plants (38,749) that survived were screened for chlorophyll and morphological mutations. Among the mutagens, SA followed by γ rays + SA and γ rays was most effective in inducing higher frequency and a broader spectrum of chlorophyll mutants. A wide range of morphological mutants affecting every growth stage was recorded with the highest frequency in 400 Gy γ rays + 0.04% SA treatment. These morphological mutants with desirable agronomic traits represent a valuable genetic resource for future breeding programs. This study revealed the potency of γ rays and SA in increasing genetic diversity and demonstrated the successful conduct of induced mutagenesis in the cowpea.

## 1. Introduction

In the present era of climate change and extreme weather events such as erratic rainfall, depleting land and water resources poses a significant risk to agricultural productivity. Besides climate change, a rapidly growing population expected to rise to 9.6 billion by 2050 imposes huge pressure on agriculture and allied sectors. Under these circumstances, scientists are deeply concerned with global food security, as a 70% increase in agriculture productivity would be required to meet the food and nutritional demands of the sky-high population [1]. The Food and Agriculture Organization (FAO) asserts that new plant breeding technologies must be adopted to develop staple food crops with higher yielding potential and climate resilience. Among the staple foods, pulse crops belonging to the Leguminosae family harvested entirely for their protein- and micronutrient-rich grains, are vital constituents in human diets. Therefore, pulses are ideal crops to meet the food and nutritional demands of a rapidly growing population and Sustainable Development Goal 2, which aims to achieve “zero hunger” [1,2]. At present, worldwide attention on the role of pulse crops in eradicating the hidden hunger prevalent in developing countries is greater than ever. Among the ten primary pulse crops recognized by the FAO, cowpea (*Vigna unguiculata* (L.) Walp.) is an important pulse crop based on its nutritional value and other desired qualities [3]. Cowpea is a highly nutritious warm-season legume with wide adaptability to dry ecologies of tropical and subtropical regions worldwide [4]. The green pods are a rich source of dietary protein for humans, and the leaves are used as livestock fodder [4]. Cowpea can withstand heat and drought stress, but it is susceptible to frost [5]. Worldwide, 14.4 million hectares of land are dedicated to cowpea cultivation, producing about 8.9 million tonnes [6]. Africa is the major cowpea growing region, contributing 95% of total production (Appendix A). Compared to other cowpea-producing nations, the annual mean yield of Indian cowpea is substantially low due to the lack of high-yielding varieties [3]. Therefore, an organized breeding approach is mandatory to improve yield and yield-attributing traits in cowpea. Among different breeding approaches, mutation breeding is a feasible, efficient, stout, and articulate tool to assist in creating varieties with enhanced yielding potential [7]. Therefore, a multiyear mutation breeding program was conducted in cowpea varieties using γ rays and SA as mutagens based on their high effectiveness and efficiency. The γ rays, while interacting with plant tissues, cause the radiolysis of water, which leads to the formation of free radicals. These highly reactive free radicals disrupt DNA–DNA cross-links, leading to the induction of random mutations [8]. However, the mutagenicity of SA is mediated via the synthesis of an organic metabolite, β-azidoalanine moiety (N3–CH2–CH(–NH2)–COOH), which interacts with DNA and induces AT→GC base pair transition and transversion [9,10].

The mutation breeding outcome depends on mutagenic effectiveness, efficiency, plant material, mutagen dose, and duration [11]. Therefore, it is imperative to evaluate mutagenic effectiveness and efficiency at the beginning of a multi-year mutagenesis experiment [12]. Determining mutagenic effectiveness and efficiency would reveal the optimum dose of mutagens. The optimum mutagen doses have successfully developed and officially released several hundred improved mutant varieties [13]. Sixteen mutant varieties of cowpea with improved agronomic traits have been developed; however, a variable range of mutagen doses have been employed in different cowpea genotypes [13]. For instance, in India, 100 to 300 Gy γ rays, in Costa Rica, 100 Gy γ rays, in Zimbabwe, 150 Gy γ rays were employed to develop different cowpea mutant varieties with improved grain yield, fodder, stress tolerance, and other agronomic traits [14]. It is evident from the literature that a variable range of γ rays has been used to enhance yield without a shred of conclusive evidence on the optimum dose. Gamma rays are the most effective and efficient mutagen in developing several cowpea mutants. In contrast, SA and combined mutagens were not successful in developing a single cowpea mutant variety. Considering the necessity of the genetic improvement of cowpea and evaluating the comparative effectiveness and efficiency of the single and combination treatments of γ rays and SA, the present study of induced mutagenesis was undertaken under field conditions.

## 2. Results

### 2.1. Seed Germination

In M_2_ generation, non-significant and significant decreases in seed germination were recorded at lower and higher mutagen doses, respectively. In the var. Gomati VU-89, seed germination was recorded in the untreated population as 93.33%, and it reduced non significantly from 88.33 to 80.67% in G1–G4 treatments, 87.33 to 81.33% in S1–S4 treatments, and 87.00 to 77.67% in G1S1–G4S4 treatments (Figure 1).

In the var. Pusa-578, a significant decrease in the seed germination was recorded in G2, G3, G4, S3, S4, G2S2, G3S3, and G4S4 mutagen doses. However, a nonsignificant decrease in seed germination was recorded in G1, S1, S2, and G1S1 mutagen doses. In the var. Pusa-578, seed germination was recorded in the untreated population as 92.00%. It decreased from 85.00 to 80.33% in G1–G4 treatments, 88.33 to 76.33% in S1–S4 treatments, and 83.67 to 75.33% in G1S1–G4S4 treatments (Figure 2).

### 2.2. Pollen Fertility

In the present study, we recorded inconsistent pollen fertility in various mutagen doses of the M_2_ generation. In the var. Gomati VU-89, non-significant and significant decreases in pollen fertility were recorded at lower (G1, G2, G3, S1, S2, S3, G1S1, G2S2, and G3S3) and higher mutagen doses (G4, S4, and G4S4), respectively. In contrast, non-significant and significant decreases in pollen fertility at lower (G1, G2, S1, S2, G1S1, and G2S2) and higher mutagen doses (G3, G4, S3, S4, G3S3, and G4S4), respectively, were recorded in the var. Pusa-578. The pollen fertility reduction ranged from 6.41 to 15.90% in G1–G4 treatments, 5.90 to 20.00% in S1–S4 treatments, and 5.13–22.82% in G1S1–G4S4 treatments in the var. Gomati VU-89 (Figure 3), while it ranged from 9.87 to 21.01% in G1–G4 treatments, 8.86 to 23.29% in S1–S4 treatments, and 10.89 to 26.33% in G1S1–G4S4 treatments in the var. Pusa-578 (Figure 4).

### 2.3. Chlorophyll Mutants: Frequency and Spectrum

Chlorophyll mutations helped us to visualize mutagenic potency and induced genetic alterations in cowpea progenies. The number of M_1_ plant progenies, segregating progenies, and percent of mutated plants (Mp) were recorded in both varieties (Table 1). A broad spectrum of chlorophyll mutants such as *Albina*, *Chlorina*, *Xantha*, *Tigrina*, *Viridis*, and *Xanthaviridis* were observed in the M_2_ generation (Figure 5) with the following description:*Albina*: Lethal mutants lacking all photopigments, leaves were white. Seedlings survived for two weeks after germination.*Chlorina*: The first pair of seedling leaves were light green. Plants remained light green throughout the growth period. Seedlings survived to maturity.*Xantha*: Leaves were yellow due to the absence of chlorophyll. Seedlings survived up to three–four leaf stages.*Tigrina*: Leaves showed yellow and green patches. Seedlings survived for 3–4 weeks.*Viridis*: Leaves were initially light/yellow-green (viridine green) but gradually turned green. Seedlings were short in height and slow-growing. Seedlings survived to maturity.*Xanthaviridis*: Leaves were a viridine green color. Seedlings survived to maturity.

A progressive increase in the frequency of chlorophyll mutants was recorded in the lower and intermediate treatments of γ rays and SA (Table 2). Moreover, combination treatments significantly increased mutation frequency compared to individual treatments in both varieties. The frequency of chlorophyll mutants ranged from 0.72 to 1.06% in G1–G4 treatments, 0.88 to 0.92% in S1–S4 treatments, and 0.53 to 1.68% in G1S1–G4S4 treatments in var. Gomati VU-89. While in the var. Pusa-578, it ranged from 0.58 to 0.77% in G1–G4 treatments, 0.89 to 1.02% in S1–S4 treatment and 0.48 to 1.78% in G1S1–G4S4 treatments. The estimation of the interaction coefficient (k) revealed the synergistic effects of mutagens (Table 2). The coefficient of interaction (k) values were significant at P = 5% based on Duncan’s multiple range test in both varieties.

The overall spectrum of chlorophyll mutants was *Albina* > *Chlorina* > *Xantha* > *Viridis* > *Xanthaviridis* > *Tigrina* in the var. Gomati VU-89, and *Albina* > *Chlorina* > *Viridis* > *Xantha* > *Tigrina* > *Xanthaviridis* in the var. Pusa-578 (Table 3). Among the doses, the frequency of *Albina*, *Chlorina*, and *Tigrina* mutants were non significantly higher in γ rays, SA, and combined treatments, respectively (Appendix A).

### 2.4. Mutagenic Effectiveness and Efficiency

The mutagenic effectiveness and efficiency were maximum at lower and intermediate doses of combined and individual mutagen treatments, respectively (Table 4). In both varieties, the effectiveness of mutagen treatments decreased progressively with the increase in mutagen doses. Mutagenic effectiveness was highest in SA, followed by γ rays + SA and γ rays. The effectiveness of SA was as high as 41.66% compared to 0.06% for γ rays. In the γ rays-treated population, the non-significant effectiveness ranged from 0.02 to 0.06 (Gomati VU-89) and 0.01 to 0.03 (Pusa-578). In the SA-treated population, significant effectiveness ranged from 25.00 to 41.66 (Gomati VU-89) and 16.66 to 41.66 (Pusa-578). In γ rays + SA-treated population, non-significant effectiveness ranged from 0.10 to 1.16 (Gomati VU-89) and 0.06 to 1.00 (Pusa-578). Mutagenic efficiency was determined based on the frequency of mutation and the extent of biological damage, viz., seedling injury (Mp/I), pollen sterility (Mp/S), and meiotic aberrations (Mp/Me). In the case of γ rays, G2 treatment, in case of SA, S1 treatment, and in case of combined mutagen, G1S1 treatment proved to be the most efficient concentrations derived from seedling injury, pollen sterility, and meiotic anomalies in both the varieties (Table 4). A progressive decrease in mutagenic effectiveness was recorded beyond the dose of G2, S2, and G1S1 with an increase in dose/concentration.

### 2.5. Morphological Mutations

Mutations affecting gross morphological traits such as plant size (height), growth habits, leaves, flowers, pods, and seeds were considered viable mutations in M_2_ generation. These mutants were named based on characters continuously detected during the study. The significant maximum frequency of morphological mutants was recorded in G2S2 in Gomati VU-89 (2.18%) and Pusa-578 (2.41%) (Table 5).

The combined mutagen doses induced the highest number of morphological mutants with no additive effects (based on k values) in both varieties (Table 6 and Table 7). The maximum significant frequency of morphological mutants was associated with the seeds, followed by flowers, growth habits, plant size (height), leaves, and pods in both the varieties (Appendix A; Table 6 and Table 7).

#### 2.5.1. Plant Height Mutants

Control: The height in untreated plants was 180.32–182.61 cm (Figure 6a).Tall mutants: These were tall with broader, dark green foliage, sparse branching, extended internodes, normal seed set, and attained a height of 180–185 cm (Figure 6b). These were induced at a frequency of 0.11 and 0.09% in the G1- and S2-treated populations of varieties Gomati VU-89 and Pusa-578, respectively.Dwarf mutants: These mutants remained dwarf throughout the growth period, exhibited short internodes, few leaves, reduced pod and seed size, low yield, and were severely stunted, measuring 110–115 cm in height (Figure 6c). These were induced at a frequency of 0.04% in the G2- and S3-treated populations of varieties Gomati VU-89 and Pusa-578, respectively.Semi-dwarf mutants: These mutants showed a height of 140–150 cm, shorter internodes, decreased branches, pods, and yield (Figure 6d). These were induced at a frequency of 0.01% in the G2S2- and S2-treated populations of varieties Gomati VU-89 and Pusa-578, respectively.

#### 2.5.2. Growth Habit Mutants

Control: Cowpea is an annual herb with a glabrous stem and a robust taproot system with erect or climbing growth habits.Semi-dwarf spreading mutants: These mutants reflected Gigas-like characteristics, vigorous growth, longer internodes, broader leaves, and spreading branches with wide branch angles (Figure 6e). These were isolated at a frequency of 0.03 and 0.02% in the G2- and S3-treated populations of varieties Gomati VU-89 and Pusa-578, respectively.Bushy mutants: These were short with condensed internodes, compact branches, and leaflets (Figure 6f). These mutants appeared at a frequency of 0.08% and 0.07% in the G3- and S2-treated populations of Gomati VU-89 and Pusa-578, respectively.One-sided branching mutants: These mutant plants showed branches on one side of the stem, a few pods, and shriveled seeds. These mutants appeared at a frequency of 0.03 and 0.02% in the G2S2- and S3-treated populations of Gomati VU-89 and Pusa-578, respectively.Axillary branched mutants: These were profusely branched with reduced internodes and yield (Figure 6g). These were induced at a frequency of 0.03 and 0.02% in the G1- and S2-treated populations of varieties Gomati VU-89 and Pusa-578, respectively.Prostrate mutants: These were initially straight but showed a trailing habit at the soil surface due to vigorous secondary branching (Figure 6h). These mutants were induced at a frequency of 0.02% and 0.03% in the G1S1- and G3S3-treated populations of varieties Gomati VU-89 and Pusa-578, respectively.

#### 2.5.3. Leaf Mutants

Control: The first pair of true leaves are simple and opposite. The leaves are dark, green, compound, smooth, dull to shiny, and pubescent with three oval leaflets. The two side leaflets are asymmetrical, and one central terminal leaflet is symmetrical (Figure 7a).Broad-leaved/Gigas mutants: These mutants were tall, with profuse secondary branching with broader leaflets, extended rachis, and robust growth. The leaflets were larger (two times bigger than the control) with broad lamina. These mutants were induced in moderate γ rays and SA treatments with a frequency of 0.06 and 0.05% in the G1S1- and G3S3-treated populations of varieties Gomati VU-89 and Pusa-578, respectively (Figure 7b).Narrow-leaved mutants: These mutants possessed narrow or small needle-like leaflets, pointed leaf tips, small pods, and few seeds. Branching was normal; however, flowering and maturity were delayed (Figure 7c). Such mutants appeared at a frequency of 0.05% and 0.04% in the G2S2- and S3-treated populations of Gomati VU-89 and Pusa-578, respectively.Elongated rachis: These mutants revealed increased rachis, narrow leaflets, and pointed leaf tips (Figure 7d). Such mutants appeared at a frequency of 0.04 and 0.03% in the G1- and G3S3-treated populations of varieties Gomati VU-89 and Pusa-578, respectively.Altered leaf architecture: These mutants exhibited notched leaflets, irregular leaf margins, abnormal leaf tips, and venation (Figure 7e,f). These mutants appeared at a frequency of 0.03 and 0.02% in the G2- and S3-treated populations of Gomati VU-89 and Pusa-578, respectively.Abnormal leaflet number: These plants were characterized by an abnormal number of leaflets, including two (bifoliate), four (tetrafoliate), and five (pentafoliate) mutants (Figure 7g–l). These mutants were not stable and showed segregation in the subsequent generations. Hence, these mutants were not included in the frequency calculations.

#### 2.5.4. Flower Mutants

Control: The flowers were usually in pairs, yellowish in color, with racemose inflorescences born on peduncles in the leaf axils. Peduncles were 2–20 cm long, stout, and grooved. The flowers were 2–3 cm in diameter with a straight keel, diadelphous stamens, a sessile ovary with several ovules, a bearded style, and an oblique stigma (Figure 8a).Flower color mutants: These were characterized by flowers that gradually turned white, blue, and red instead of yellow in the parent variety. Few flower mutants exhibited variation even in the color of the petals (Figure 8b). Blue flowers were recorded more frequently than white flowers in Gomati VU-89. However, white mutants appeared more often than blue flowers in Pusa-578 (Figure 8c). The G3 treatment showed a higher frequency of flower mutants in Gomati VU-89 (0.03%) and Pusa-578 (0.04%).Multiple flower mutants: In these mutants, each peduncle consisted of three to four normal flowers instead of two flowers in the parent variety. These mutants appeared at a frequency of 0.03% in the G1S1- and S1-treated populations of varieties Gomati VU-89 and Pusa-578, respectively (Figure 8d,e).Open flower mutants: These possessed flowers with broad keel and wings, exposed stamens, and stigma. Such mutants appeared at a frequency of 0.06 in the G3- and G4S4-treated populations of varieties Gomati VU-89 and Pusa-578, respectively (Figure 8f).Non-flowering mutants: These mutants did not flower at all and remained vegetative throughout the growth period. These appeared at a frequency of 0.01 and 0.02% in the G4- and G4S4-treated populations of varieties Gomati VU-89 and Pusa-578, respectively.Late flowering mutants: In these mutants, the flowering was delayed by 9 to 10 days compared to untreated plants. Such mutants were commonly observed in populations treated with higher mutagen doses at a frequency of 0.04% and 0.03% in the G4S4- and G4-treated populations of varieties Gomati VU-89 and Pusa-578, respectively.Early maturing mutants: These mutants matured 3 to 4 days earlier than the untreated population and appeared at a frequency of 0.02% in the G2- and S2-treated populations of varieties Gomati VU-89 and Pusa-578, respectively.

#### 2.5.5. Pod Mutants

Control: Pods were pending and vertical, 23–30 cm long, 5–10 mm wide, containing 9–12 seeds. Pods occurred singly or in pairs (Figure 9a).Small/narrow pods: These mutants possessed narrow and small pods that appeared at a frequency of 0.07 and 0.06% in the G2S2- and S3-treated populations of varieties Gomati VU-89 and Pusa-578, respectively (Figure 9b).Bold-seeded pods: These mutants showed robust growth, profuse branching, broad leaflets, large-sized flowers, and longer pods containing bold seeds. Such mutants were induced at lower and intermediate doses with 0.06 and 0.05% frequency in the G1- and S2-treated populations of Gomati VU-89 and Pusa-578, respectively (Figure 9c).Pod width and length mutants: These mutants exhibited variations in pod attributes such as pod width (Figure 9d–e) and pod length (Figure 9f). However, such mutants were not stable and showed segregation in the subsequent generations and hence were not included in frequency calculations.

#### 2.5.6. Seed Mutants

Control: Seeds were brown and white with smooth seed coats in Gomati VU-89 and Pusa-578, respectively (Figure 10).Seed coat color mutants: These were upright and straight, with light green leaves, compared to the parent variety’s dark green leaves. Such mutants were characterized by red or black smooth seed coats. Such mutants were induced at an equal frequency of 0.06% in the G2S2- and G3S3-treated populations of Gomati VU-89 and Pusa-578, respectively.Seed coat pattern mutants: These mutants revealed streaked, speckled, and stippled rough seed coats. Such mutants appeared at a frequency of 0.07% and 0.06% in the G4- and S3-treated populations of Gomati VU-89 and Pusa-578, respectively.Seed shape and surface mutants: These mutants revealed alterations in seed attributes and appeared at a frequency of 0.09% and 0.08% in the G2S2- and S3-treated populations of varieties Gomati VU-89 and Pusa-578, respectively.

## 3. Discussion

### 3.1. Seed Germination

Evaluating the mutagenic sensitivity of germinating seeds constitutes a critical aspect of mutagenesis, as entire plant growth and development depend upon the seedling establishment. The results revealed a dose-dependent increase in germination inhibition percent in both varieties. Among mutagens, inhibition was maximum in combined mutagen doses, which could be attributed to the synergistic effect of the mutagens. The results were in good agreement with the previous findings of Raina et al. [15] that reported reduced germination in cowpea seeds treated with γ rays and SA. The inhibition in germination may be attributed to the metabolic disturbances at the cellular level, chromosomal damage, and the reduced activity of phytohormones following the mutagen treatment [16].

### 3.2. Pollen Fertility

In the present study, increased pollen sterility in plants raised from mutagen-treated seeds could be due to mutagen-induced chromosomal aberrations and physiological and genetic changes resulting in aberrant pollen grains [16,17]. The results were also supported by earlier studies that reported mutagen-induced pollen sterility in crops such as *Lablab purpureus* [18], *Lens culinaris* [19,20], *Macrotyloma uniflorum* [21], and *Cajanus cajan* [22]. The pollen sterility percent was remarkably less in M_2_ than in the M_1_ generation, indicating a recovery operation between generations.

### 3.3. Chlorophyll Mutations

In mutagenesis, the induction of chlorophyll mutants is useful in assessing the genetic effects and sensitivity of different mutagens [23]. Due to its better precision in scoring, the frequency of chlorophyll mutants is one of the reliable indices for assessing a mutagen’s power, capacity, influence, efficacy, and potency [24,25,26,27]. Regardless of its negative impact on the early growth stages, chlorophyll mutants are important in mutagenesis. The present study recorded the maximum frequency of chlorophyll mutations in seedlings raised from seeds treated with intermediate doses. Pawar et al. [18] also reported that the intermediate treatment of gamma rays (0.500 KR) and EMS (0.3%) showed a higher frequency of chlorophyll mutations in *Zingiber officinale*.

On the contrary, Singh et al. [28] reported a higher frequency of chlorophyll mutants in cowpea treated with higher doses of γ rays. Goyal and Khan [29] also reported a higher incidence of chlorophyll mutants with increased γ ray doses. In the present study, decreased chlorophyll mutants at higher mutagen doses may be due to the saturation of mutations, which leads to the exclusion of the mutant cells. Among mutant types, combined mutagens induced more frequency of *Albina* and *Chlorina* mutants than individual mutagen doses, which may be due to the synergistic effects of the mutagens. The synergism between combined mutagen doses has also been reported earlier in *Cicer arietinum* [30] and *Linum usitatissimum* [31]. The possible reason for synergism might be that the first mutagen may have exposed the accessible protected mutable sites to the second mutagen that rendered the repair enzymes non-functional, indirectly facilitating the mutation fixation induced by the former mutagen [32].

### 3.4. Mutagenic Effectiveness and Efficiency

The evaluation of effectiveness and efficiency is essential in determining the usefulness of mutagens in inducing beneficial mutations and the selection of mutants with desired traits [24]. For an effective selection, the mutation treatment should not induce biological damages such as chromosomal anomalies and bio-physiological and genotoxic effects, which diminish cell survival and eventually eradicate the mutation. Effectiveness indicates mutations induced by a unit dose of mutagen [33,34]. On the contrary, mutagenic efficiency refers to the mutation frequency in relation to biological damages such as seedling injury, pollen sterility, and meiotic abnormalities induced in the M_1_ generation. Therefore, effectiveness reflects genotypic sensitivity, and efficiency indicates mutagenic potency.

In the present study, mutagenic effectiveness was minimum at higher treatments of combined and individual mutagens, which might be attributed to more prominent mutation effects in the cell that led to meiotic anomalies, bio-physiological alterations, and reduced cell survival. This is in propinquity with the results obtained by Bhosale and Kothekar [34] in *Cyamopsis tetragonoloba* treated with 5 to 15 kR gamma rays. Several workers reported that the decreased effectiveness of higher mutagen doses could be attributed to the failure of a proportional increase in mutation frequency with the increase in dose/concentration [35].

A highly effective mutagen may not inevitably show higher efficiency and vice versa. Mutagenic efficiency depends on multiple factors such as mutagenic reactivity with the material, its applicability to the biological system, and the degree to which biological damage is induced [36]. The higher efficiency reflects comparatively less biological damage (seedling injury, pollen sterility, meiotic anomalies) in relation to induced mutations. The efficiency of the mutagens was evaluated based on criteria such as seedling injury (Mp/I), pollen sterility (Mp/S), and meiotic anomalies (Mp/Me). Each criterion showed ample variations in the values of efficiencies for the same mutagen dose, indicating the elasticity of using one or all mutagens at a time.

In the present study, efficiency was maximum based on meiotic anomalies, which may be more due to the low frequency of chromosomal aberrations than seedling injury and pollen sterility. Among the mutagens, efficiency was maximum in combined mutagens, which may be attributed to the highest frequency of chlorophyll mutations. However, lower and intermediate treatments revealed enhanced efficiency among the mutagen treatments. The higher effectiveness and efficiency of lower or intermediate treatments of γ rays and chemical mutagens have been confirmed earlier in lentil [37], cowpea [38], and chickpea [39]. The higher efficiency of lower and intermediate doses may be ascribed to the progressive increase in biological damage with an increase in dose at a rate higher than the frequency of mutations [33].

### 3.5. Morphological Mutations

Several plants exhibited morphological deviations in plant height, growth habits, leaf, flower, pod, and seed attributes were recorded in the present study. The genetic diversity evaluated in 46 morphological mutants included tall/dwarf plant heights, growth habits, leaf shapes, flower color, early/late maturity, pod shape, seed coat, and texture. The morphological mutations affecting a single character are attributed to changes in single genes (monogenic) [40]. In contrast, morphological mutations affecting more than one character are attributed to the mutation of the pleiotropic gene, altered gene clusters, and chromosomal breakage [41,42,43,44]. In this study, we followed the mutation classification of Gnanamurthy et al. [45] and defined macro mutations as qualitatively inherited, and morphologically distinct alterations included flower and seed coat color mutations. In contrast, micro mutations were described as quantitatively inherited, and phenotypically invisible alterations included mutations affecting plant height, growth habits, and pod attributes. Morphological mutants may be ascribed to a mutation in the genes governing the ontogeny of organs through their gene products. In the present study, gamma rays induced DNA breaks, chromosomal anomalies, altered auxin metabolism, mineral, amino acids, and physio-morphogenetic variations in cells, which may be attributed to morphological mutations [46,47,48]. Besides gamma rays, sodium azide-induced point mutations, rarely small deletions, and other chromosomal rearrangements may also influence plant growth and development [49,50,51]. Therefore, isolated morphological mutants might possess point mutations rather than small deletions. This could be due to the rigorous selection procedure applied in the M_1_ generation, where mutants with gross morphological and chromosomal anomalies were rejected in each treatment. Only normal-looking plants were advanced to subsequent generations. However, whether the morphological mutants were due to mutations in single genes with pleiotropic effect or multiple genes is not clear. Mostly, morphological mutations were nonsegregating in the later generations of the present study and could represent valuable genetic resources in the future breeding programs in the following manner:

The plant-height mutants (dwarf and semi-dwarf) isolated in the present study were resistant to lodging due to short basal branches. Badigannavar and Mondal [52] also reported dwarf, semi-dwarf, and tall mutants in groundnut treated with 150 Gy, 250 Gy, and 350 Gy gamma rays. A maximum number of tall and dwarf mutants were reported in cowpea treated with 25 mM ethyl methanesulphonate [53]. Anjana and Thimmaiah [53] also reported dwarf mutants in cowpea using γ rays. The plant-height mutants may be attributed to the mutagen-induced variations in the expression of genes (GA20 oxidase) governing gibberellic acid synthesis [54]. These plant-height mutants, particularly dwarf ones, were lodging resistant and had increased root nodules, thereby having a higher nitrogen-fixing ability that could be exploited in future breeding programs.

In the present study, a broad spectrum of mutations affecting growth habits such as bushy and spreading mutants, important from the breeder’s perspective, was isolated. According to the previous studies, bushy growth habits were attributed to a mutagen-induced increase in lateral branches and higher photosynthetic activities in the mungbean [55]. On the contrary, Horn et al. [4] isolated bushy growth habit mutants in γ-irradiated cowpea with fewer branches. Hall [56] and Martins et al. [57] opined that bushy growth habits might be attributed to the plant’s altered physiological properties, including leaf senescence and indeterminate growth habit. Similarly, spreading growth habit mutants were isolated in cowpea treated with different doses of γ rays [4]. Singh et al. [28] also reported spreading, and semi-spreading cowpea types yielded less grain and more fodder. The one-sided branching mutant isolated in the present study may be attributed to mutagen-induced altered hormone synthesis [58] and branching patterns [59].

Flower mutants were observed in almost all the mutagenic treatments in both varieties. Mutagen-induced variations in floral abnormalities have been reported in various crops such as lentil [35], faba bean [51], and cowpea [53]. The lower and intermediate doses of γ rays and SA positively affected flowering days, and some of the mutant lines flowered 3 to 4 days earlier than the control. However, late-flowering mutants raised from seeds treated with higher doses of mutagens flowered 9 to 10 days later than the control. Horn et al. [4] also reported late-flowering cowpea mutants irradiated with 300 Gy γ rays. Maluszynski et al. [25] opined that late-flowering mutants were common in plants treated with higher doses of mutagens. Both early and late flowering mutants are important from the breeder’s perspective and offer choices to breeders from a more extensive breeding stock for various breeding traits and purposes [60]. Adekola and Oluleye [61] reported early maturing mutants in cowpea var. IT 84S2246D treated with 196 and 245 Gy γ rays. Early maturing mutants possess multiple advantages over the parent variety; these include the ability to escape or tolerate insect damage and prevent insect populations from building up due to the short duration of the reproductive phase [62]. A reduction in the maturity period would be required to escape the heat stress as cowpea is mainly grown as a summer-season crop and faces heat stress during its pod-filling stage. Early maturing mutants also possess better tolerance to drought and thrive in areas receiving less rainfall. Flower mutations could be attributed to the mutagen-induced physiological or biochemical alterations in the differentiation of floral whorls [63]. The open flower mutants (with exposed stamens and stigma) could be helpful in open pollination and establish an alternate mechanism for hybrid seed production. However, flower color mutation makes plants more attractive to foraging by insects. Such plants are more susceptible to herbivory when left unprotected [64].

A broad spectrum and higher frequency of leaf mutations were observed in lower and intermediate doses of γ rays and SA, with prominent alterations in size, shape, number, and arrangement of leaflets. The mutagen-induced alterations in leaf attributes resulted in leaf mutants such as broad-leaved, narrow-leaved, elongated rachis, and altered leaf architecture. Similar leaf mutants have been reported in several crops, such as chickpea [65], lentil [35], and faba bean [51]. Broad-leaved mutants may be more efficient and possess an advantage over the parent variety in having a high leaf area index and receiving more solar radiation that could enhance the photosynthetic rate [66]. Mutagen-induced increases in multi-foliation could increase photosynthetic activities and biomass production, positively impacting seed yield in cowpea breeding programs.

Similarly, leaf mutations such as tetrafoliates and pentafoliates were reported in cowpea treated with EMS [53] and mungbean treated with gamma rays and EMS [67]. Almost all leaf mutants (except non-flowering) showed a stable phenotype in the subsequent M_3_ generation. Leaf mutations could be attributed to mutagen-induced cellular damage, chromosomal breakage, altered mineral metabolism, and the disrupted synthesis and transport of auxin [53].

The pod mutants isolated in the present study, including bold-seeded, long, and broad pod mutants, showed luxuriant growth, increased primary and secondary branches, flowers, and seed yield. Mutagen-induced bold-seeded mutants have also been reported in *Vigna mungo* [68], *Cicer arietinum* [69], *Linum usitatissimum* [70], and *Vicia faba* [71,72]. Similarly, broad and narrow pod mutants have been reported in crops such as *Vicia faba* [51] and *Lens culinaris* [35]. The pod mutations may be attributed to mutagen-induced alterations in the pleiotropic gene, gene clusters, or chromosomal rearrangement. The bold-seeded mutants isolated in the present study showed substantial improvement in yield. They could be utilized as donor parents for the bold character or released directly as new cultivars after being subjected to multilocation trails. In the present study, longer pod mutants were high-yielding compared to the parent variety. The increased seed yield may be because longer pods contain more seeds and less biomass [60]. Horn et al. [4] reported that mutations caused increased pod length in γ-irradiated cowpea mutant lines. Other critical effects of the mutation recorded in the present study were the range of variations in seed attributes. Mutations affecting seed attributes could impact the marketability of cowpea, as seed shape, seed coat color, and texture influence consumers’ preferences. Mutational events could be attributed to variations in seed attributes [38]. Shen et al. [73] opined that mutagens altered the genes associated with seed coat color stability in Brassica rapa. Abnormal enzymatic activity in starch-branching enzyme genes influenced the seed coat texture in *Pisum sativum* [74]. Several morphological mutants with desired agronomical traits are of immense practical utility; these may be useful in gene mapping and phylogenetic studies. These may also serve as parents in crossing programs and may be released directly as improved varieties after being subjected to multilocation trials [75,76,77].

Morphological mutants showed varying frequency and a broader spectrum between and within the varieties, reflecting the differential response of varieties toward mutagen doses. The earlier workers have also reported the differences in varietal response to mutagenic doses in crops such as faba bean [51] and lentil [35]. In both varieties, combined mutagen treatments induced a maximum frequency of morphological mutations. The results were in good agreement with the findings of Laskar and Khan [37], that reported the maximum frequency of morphological mutations in combined mutagen treatments compared to individual mutagen treatments in lentils. The morphological mutants depicted in Table 6 and Table 7 were non-segregating in the subsequent generations. Several researchers have also reported non-segregating morphological mutants in advanced generations [78,79,80,81,82,83,84,85,86,87].

## 4. Materials and Methods

### 4.1. Experimental Materials and Seed Irradiation

M_0_ seeds of two cowpea varieties were obtained from the National Bureau of Plant Genetic Resources, New Delhi (Appendix A). In the beginning, a pilot experiment was carried out to optimize the mutagen doses of γ rays and SA [88]. Based on the available literature, seeds were irradiated with different doses of γ rays, viz. 100 Gy (G1), 200 Gy (G2), 300 Gy (G3), 400 Gy (G4), 500 Gy (G5), 600 Gy (G6), 700 Gy (G7), 800 Gy (G8), 900 Gy (G9), and 1000 Gy (G10) at a dose rate of 11.58 Gy/min using Gamma chamber Model-900 with Cobalt-60 radioisotope under standard conditions at the National Botanical Research Institute, Lucknow, India. For chemical treatments, seeds were presoaked in water for six hours and then treated with several doses of SA, viz. 0.01% (S1), 0.02% (S2), 0.03% (S3), 0.04% (S4), 0.05% (S5), 0.06% (S6), 0.07% (S7), 0.08% (S8), 0.09% (S9), and 0.1% (S10) for nine hours at the Mutation Breeding Laboratory, Department of Botany, Aligarh Muslim University (AMU), Aligarh, India. The SA-treated seeds were washed under tap water to eliminate any mutagen adhered to the seed surface. For combination treatments, a separate set of seeds were treated with combined γ rays + SA doses, viz., G1S1 (100 Gy γ rays + 0.01% SA), G2S2 (200 Gy γ rays + 0.02% SA), G3S3 (300 Gy γ rays + 0.03% SA), G4S4 (400 Gy γ rays + 0.04% SA), G5S5 (500 Gy γ rays + 0.05% SA), G6S6 (600 Gy γ rays + 0.06% SA), G7S7 (700 Gy γ rays + 0.07% SA), G8S8 (800 Gy γ rays + 0.08% SA), G9S9 (900 Gy γ rays + 0.09% SA), and G10S10 (1000 Gy γ rays + 0.1% SA). The study revealed that doses beyond G4 in γ ray treatments, S4 in SA treatments, and G4S4 in combination treatments were detrimental and caused more than 50% reduction in seed germination and hence were discarded [88]. Thus, seeds treated with the first four single and combined treatments were selected and advanced to subsequent generations.

### 4.2. Experimental Site and Crop Cultivation

The seeds were sown at a distance of 0.6 m (row to row) and 0.3 m (seed to seed) in the field (23.5 × 40 m) with ten blocks (each 3 × 1.8 m) of the Agricultural Faculty, AMU, Aligarh (Appendix A). All the recommended cultivation practices such as irrigation, fertilization (15–20 N, 50–60 P_2_O_5_, and 50–60 K_2_O kg/ha), and weeding were performed at regular intervals.

#### Details of the Field Trials

A total of 7800 cowpea seeds (300 seeds/treated and untreated set in each variety) were sown in the agriculture field, AMU, Aligarh, during mid-April 2014 in a randomized complete block design (RCBD). All the seeds from M_1_ plants were harvested separately, and ten healthy M_2_ seeds from each M_1_ plant were sown to raise the M_2_ generation from mid-April 2015 to October 2015. A total of 57,620 M_2_ seeds generated from the M_1_ generation of two varieties were sown in the same field to raise the M_2_ generation. A total of 47,650 seeds germinated, of which 38,749 plants survived, were screened for morphological diversity (Appendix A).

### 4.3. Field Analysis

#### 4.3.1. Seed Germination

Seed germination was recorded using the following formula:$$\mathrm{Germination}\;\left(\%\right)=\frac{\mathrm{No}.\;\mathrm{of}\;\mathrm{seeds}\;\mathrm{germinated}}{\mathrm{No}.\;\mathrm{of}\;\mathrm{seeds}\;\mathrm{sown}}\times 100$$

#### 4.3.2. Chlorophyll Mutants

Chlorophyll mutants were recorded in the M_2_ generation after 15–25 days of seed sowing. The chlorophyll mutants were categorized following the protocol proposed by Gustafsson [26]. The influence of combination doses of mutagens on the frequency of chlorophyll mutations was analyzed by calculating the coefficient of interaction (k) given by Sharma [32] and later followed by several researchers [11,35,72].
$$\mathrm{Mutation}\;\mathrm{frequency}\;\left(\%\right)=\frac{\mathrm{Number}\;\mathrm{of}\;\mathrm{mutant}\;\mathrm{seedlings}}{{\mathrm{Total}\;\mathrm{number}\;\mathrm{of}\;M}_{2}\;\mathrm{seedlings}}\times 100$$
$$\mathrm{Coefficient}\;\mathrm{of}\;\mathrm{interaction}\;\left(k\right)=\frac{\left(a\;+\;b\right)}{\left(a\right)+\left(b\right)}$$
where, (a + b) is the frequency of mutation in combined mutagen doses, (a) + (b) is the frequency of mutation in individual mutagen doses, and k is the interaction coefficient.

If k = 1, it indicates additive interaction. Any deviation from this value would show synergistic or less than additive effects.

### 4.4. Mutagenic Effectiveness and Efficiency

The mutation frequency refers to mutagenic effectiveness, while the mutation percentage and biological damage indicate mutagenic efficiency. In the M_2_ generation, mutagenic effectiveness and efficiency were calculated using the following formulae [33]:$$\mathrm{Mutagenic}\;\mathrm{effectiveness}\;\left(\gamma \;\mathrm{rays}\right)=\frac{\mathrm{Rate}\;\mathrm{of}\;\mathrm{mutation}\;\left(\mathrm{Mp}\right)}{\mathrm{Dose}\;\mathrm{in}\;\mathrm{Gray}\;\left(\mathrm{Gy}\right)}$$
$$\mathrm{Mutagenic}\;\mathrm{effectiveness}\;\left(\mathrm{SA}\right)=\frac{\mathrm{Rate}\;\mathrm{of}\;\mathrm{mutation}\;\left(\mathrm{Mp}\right)}{\mathrm{Concentration}\;\times \;\mathrm{duration}\;\mathrm{of}\;\mathrm{treatment}}$$
$$\mathrm{Mutagenic}\;\mathrm{effectiveness}\;\left(\gamma \;\mathrm{rays}+\mathrm{SA}\right)=\frac{\mathrm{Rate}\;\mathrm{of}\;\mathrm{mutation}\;\left(\mathrm{Mp}\right)\;}{\begin{array}{c} \mathrm{The}\;\mathrm{dose}\;\mathrm{of}\;\gamma \;\mathrm{rays}\;\left(\mathrm{Gy}\right)\times \\ \mathrm{the}\;\mathrm{concentration}\;\mathrm{of}\;\mathrm{SA}\;\left(\%\right)\times \\ \mathrm{duration}\;\mathrm{of}\;\mathrm{treatment} \end{array}}$$
$$\mathrm{Mutagenic}\;\mathrm{efficiency}=\frac{\mathrm{Rate}\;\mathrm{of}\;\mathrm{mutation}\;\left(\mathrm{Mp}\right)}{{*\mathrm{Biological}\;\mathrm{damage}\;\mathrm{in}\;M}_{1}\;\mathrm{generation}}$$

* Biological damage was measured based on seedling injury, pollen sterility, and meiotic abnormalities.

### 4.5. Morphological Mutants

Data on twenty-three qualitative and quantitative traits based on available cowpea descriptors from the International Union for the Protection of New Varieties of Plants [89] and the International Board for Plant Genetic Resources [90] were noted in the M_2_ generation. Morphological variations for plant height, growth habits, leaves, flowers, pods, and seeds were observed, recorded, and analyzed. The frequency (F) of the morphological mutants was calculated using the following formula:$$F\;\left(\%\right)=\frac{\mathrm{No}.\;\mathrm{of}\;\mathrm{mutants}}{\mathrm{Total}\;\mathrm{number}\;\mathrm{of}\;\mathrm{plants}\;\mathrm{studied}}\times 100$$

## 5. Conclusions

The selected lower and intermediate mutagen doses induced high frequency and a broader spectrum of morphological mutations that may be of great economic interest to cowpea breeders. Momentous variability was induced among morphological mutants using γ rays and SA. This study confirmed the potency of γ rays and SA in increasing genetic diversity and demonstrated the successful conduct of mutagenesis in the cowpea. Effectiveness and efficiency were highest at the 200 Gy γ rays, 0.02% SA, 100 Gy γ rays + 0.01% SA treatments. Hence, these doses may be employed in future breeding programs to obtain a high frequency of mutations with the least biological damage. A total of 46 morphological mutants with agronomically desirable traits are valuable genetic resources that could be exploited to enrich the genetic base of existing cowpea cultivars. Overall, the present study made extensive phenotypic selections of mutants over multiple generations and isolated elite cowpea mutants. For future research, the selected mutants with desirable traits are recommended for multilocation trails across dry agro-ecologies to ensure the fixation of new traits. The selected mutants can be used to study genes, the gene function of mutant traits, and the development of markers to facilitate marker-assisted selection.

## Figures and Tables

**Figure 1 plants-11-01322-f001:**
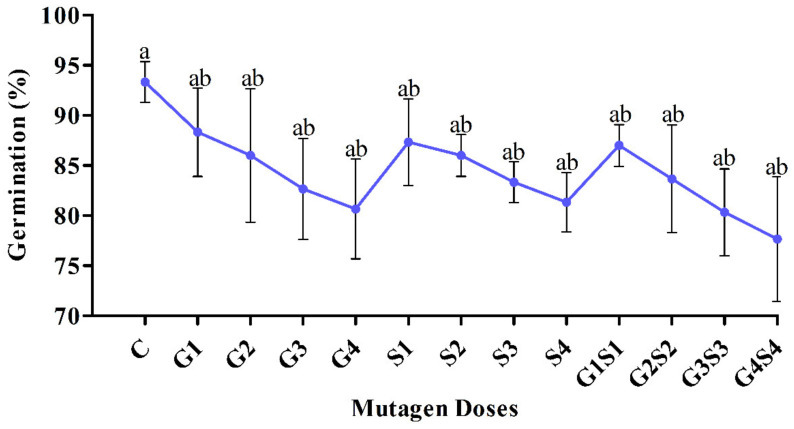
Effects of different doses of γ rays and SA on the seed germination in var. Gomati VU-89. The data is presented as percent (n = 300) and standard error. Line graphs with the same letters are not significant at a 5% level of significance, based on Duncan’s multiple range test (DMRT).

**Figure 2 plants-11-01322-f002:**
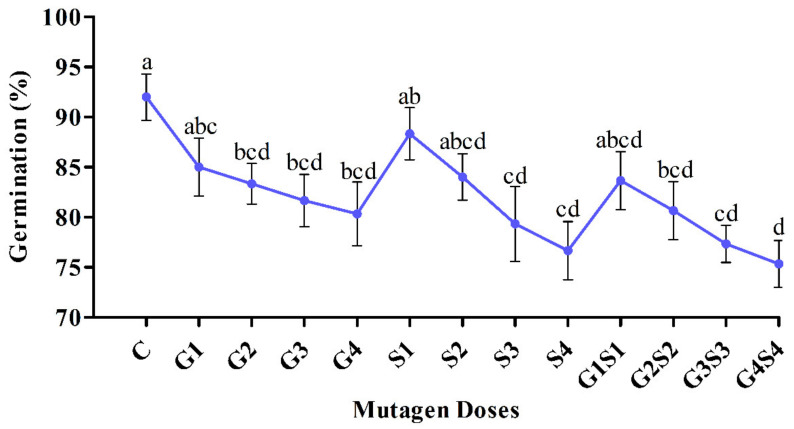
Effects of different doses of γ rays and SA on seed germination in var. Pusa-578. The data is presented as percent (n = 300) and standard error. Line graphs with the same letters are not significant at a 5% level of significance, based on Duncan’s multiple range test (DMRT).

**Figure 3 plants-11-01322-f003:**
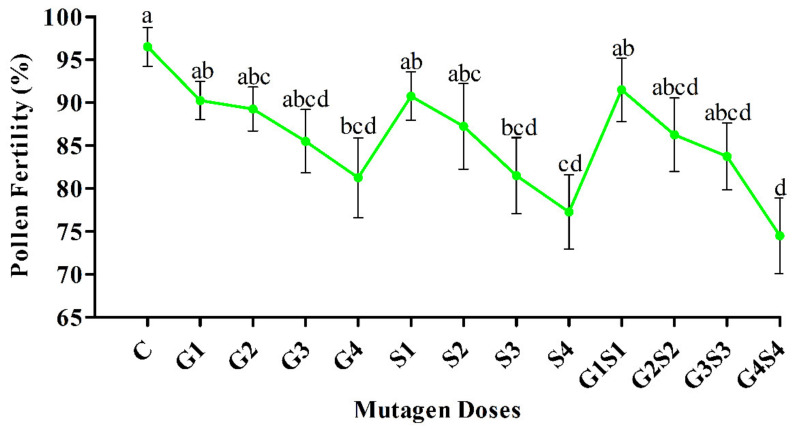
Effects of different doses of γ rays and SA on the pollen fertility in var. Gomati VU-89. The data is presented as percent (n = 400) and standard error. Line graphs with the same letters are not significant at a 5% level of significance, based on the Duncan’s multiple range test (DMRT).

**Figure 4 plants-11-01322-f004:**
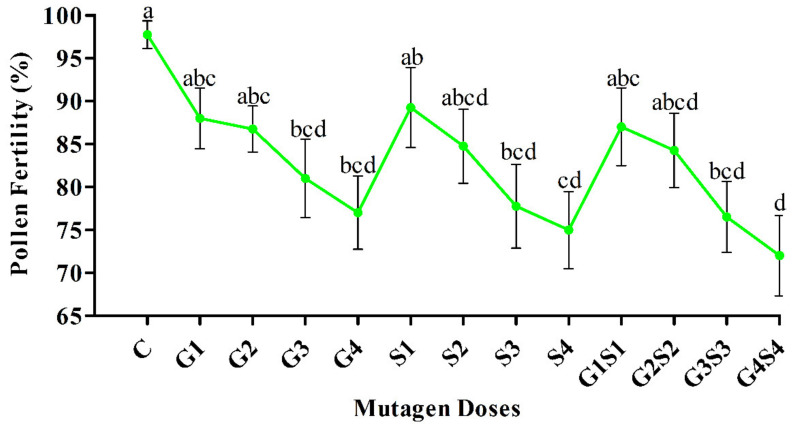
Effects of different doses of γ rays and SA on the pollen fertility in var. Pusa-578. The data is presented as percent (n = 400) and standard error. Line graphs with the same letters are not significant at a 5% level of significance, based on the Duncan’s multiple range test (DMRT).

**Figure 5 plants-11-01322-f005:**
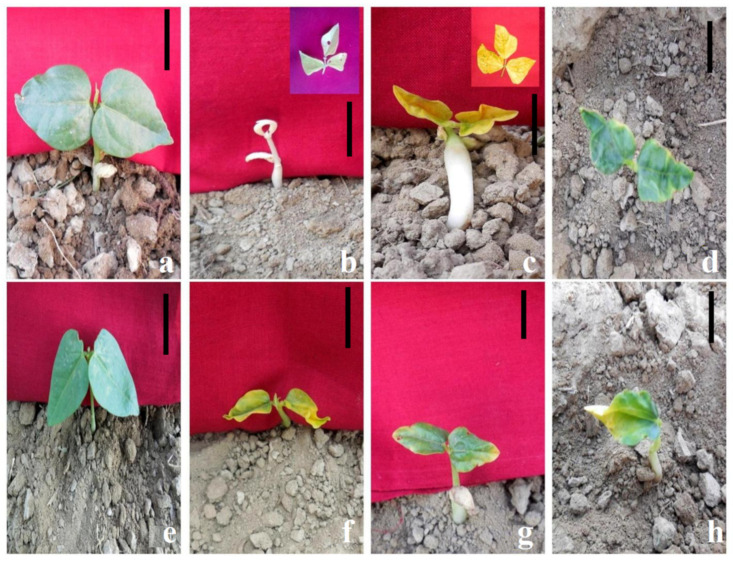
Representative photographs of γ rays and SA-induced chlorophyll mutants. (**a**). Control seedling of var. Gomati VU-89. (**b**) *Albina*, (**c**) *Xantha*, (**d**) *Tigrina*, (**e**) control seedling of var. Pusa-578, (**f**) *Chlorina*, (**g**) *Xanthaviridis*, and (**h**) *Viridis*. Scale Bar = 2 cm.

**Figure 6 plants-11-01322-f006:**
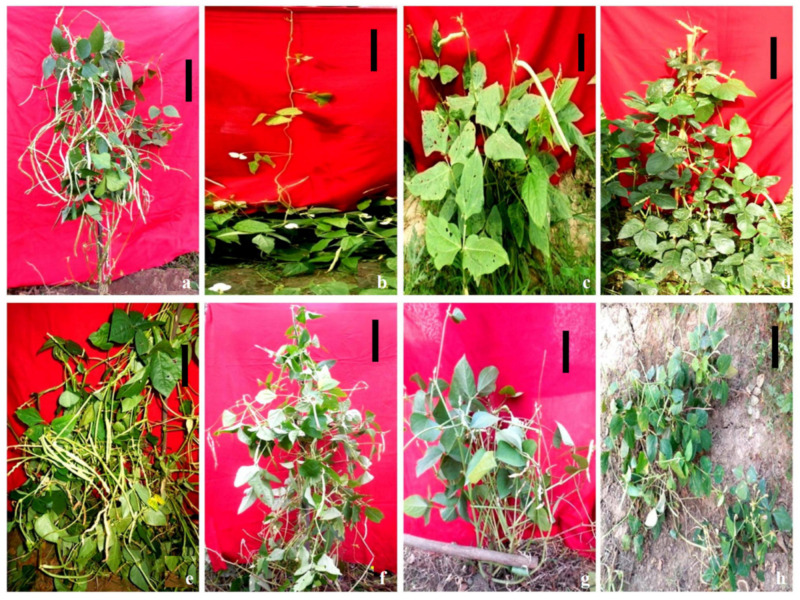
Representative photographs of gamma rays and sodium azide-induced alterations in plant height and growth habits. (**a**) Control plant, (**b**) tall mutant, (**c**) dwarf mutant, (**d**) semi-dwarf mutant, (**e**) semi-dwarf spreading mutant, (**f**) bushy mutant, (**g**) axillary branched mutant, and (**h**) prostrate mutant. Scale Bar = 30 cm.

**Figure 7 plants-11-01322-f007:**
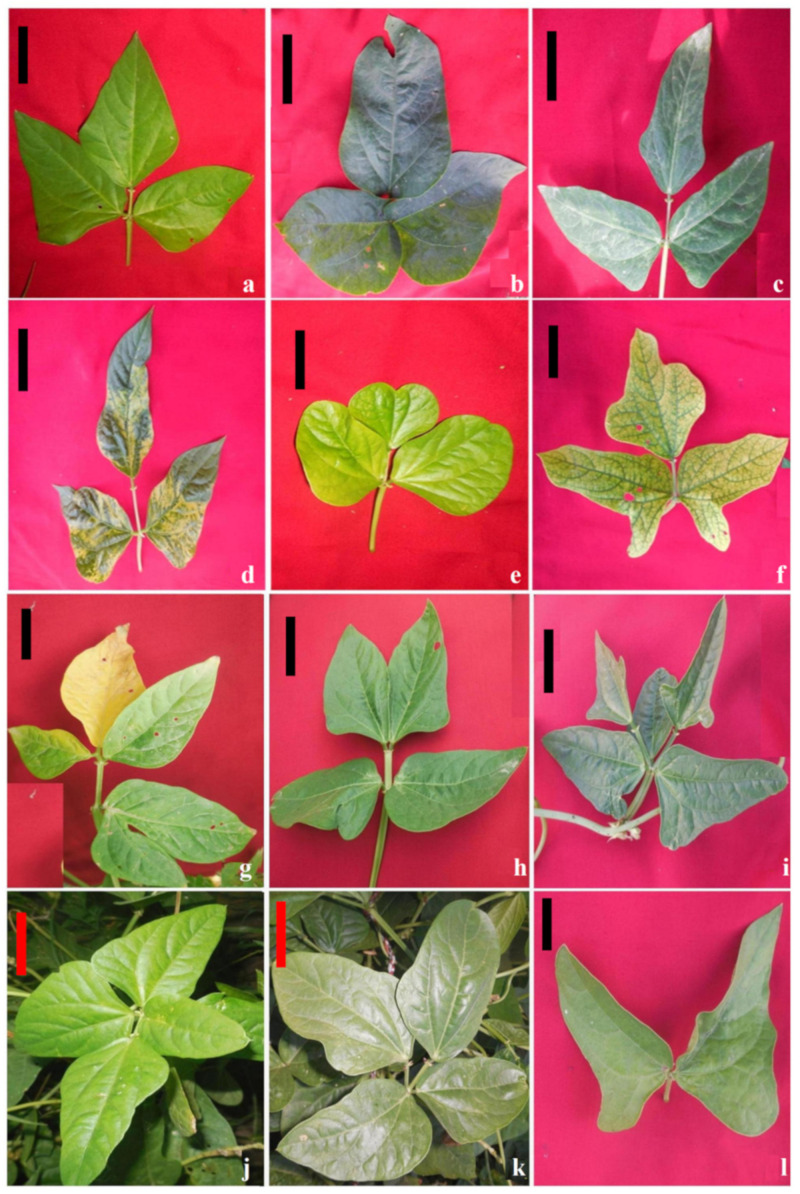
Representative photographs of gamma rays and sodium azide-induced alterations in leaf shape. (**a**) Control leaf, (**b**) broad-leaved mutant, (**c**) narrow-leaved mutant, (**d**) elongated rachis mutant, (**e**) altered leaf architecture with notched leaf tips, (**f**) altered leaf architecture with leaflet outgrowths, (**g**) mutant showing fused leaflets, (**h**) tetrafoliate with fused terminal leaflets, (**i**) pentafoliate mutant, (**j**) tetrafoliate with narrow leaflets, (**k**) tetrafoliate with broad leaflets, and (**l**) bifoliate mutant. Scale Bar = 2 cm.

**Figure 8 plants-11-01322-f008:**
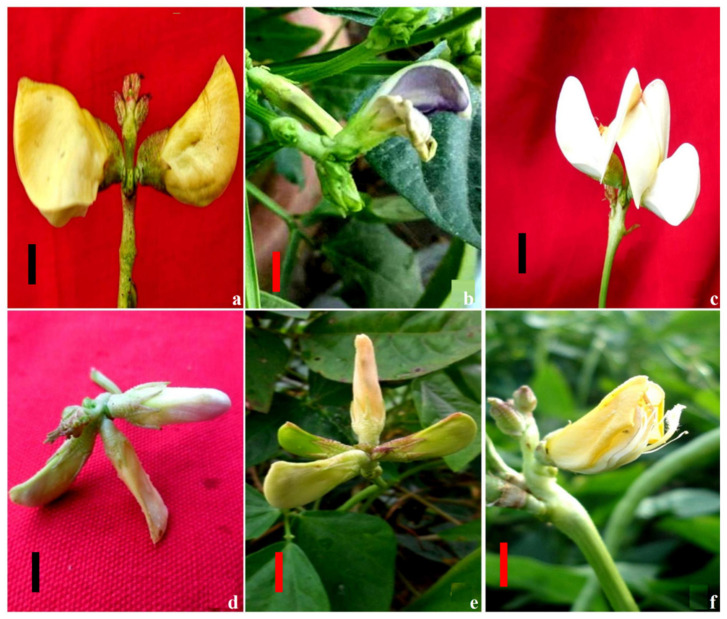
Representative photographs of gamma rays and sodium azide-induced alterations in the number and color of flowers per peduncle. (**a**) Yellow flowers in control, (**b**) blue-colored petals, (**c**) white flowers, (**d**) three flowers per peduncle, (**e**) four flowers per peduncle, and (**f**) open flower. Scale Bar = 1 cm.

**Figure 9 plants-11-01322-f009:**
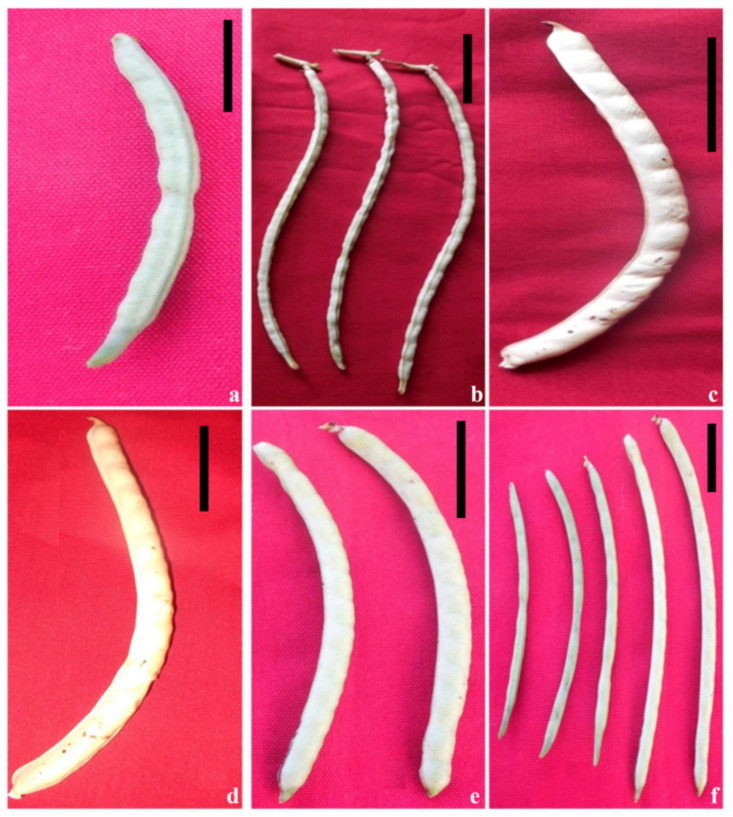
Representative photographs of gamma rays and sodium azide-induced alterations in pod size. (**a**) Control pod, (**b**) narrow pods, (**c**) bold-seeded pods, (**d**) broad pod, (**e**) pod width variations, and (**f**) pod length variations. Scale Bar = 5 cm.

**Figure 10 plants-11-01322-f010:**
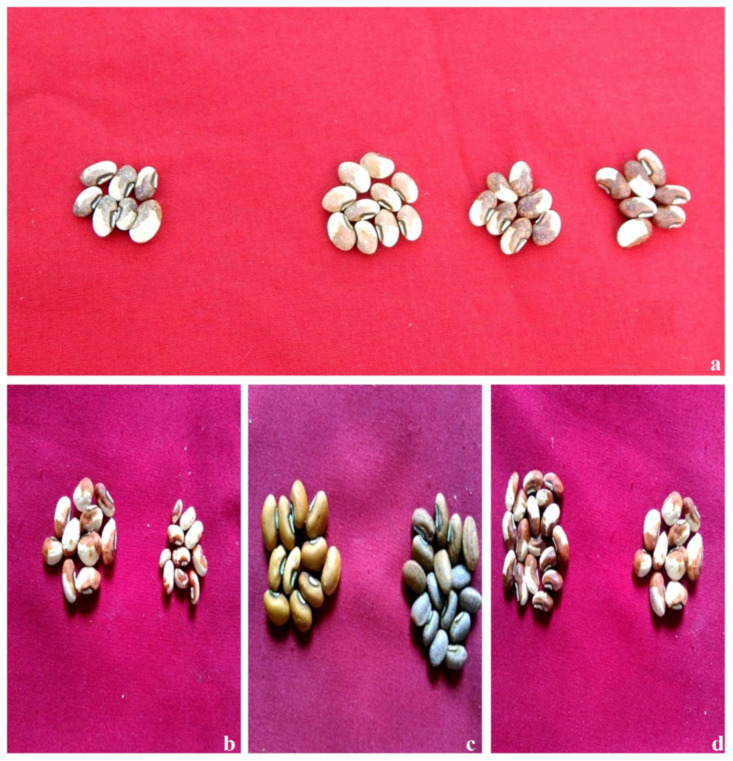
Representative photographs of gamma rays and sodium azide -induced alterations in seed pattern, color, and shape. (**a**) Seed coat color variations, (**b**) bold and small seeds, (**c**) smooth seeds, (**d**) wrinkled seeds.

**Table 1 plants-11-01322-t001:** The number of M_1_ plant progenies, plant progenies segregating in M_2_, and percent of mutated plants.

Treatment	Var. Gomati VU-89			Var. Pusa-578		
	No. of M_1_ Plant Progenies	No. of Plant Progenies Segregating in M_2_	Mutated Plant (%)	No. of M_1_ Plant Progenies	No. of Plant Progenies Segregating in M_2_	Mutated Plant (%)
C	270	0	0.00 ^f^	260	0	0.00 ^i^
G1	245	2	0.82 ^h^	235	1	0.43 ^h^
G2	223	11	4.93 ^b^	213	5	2.35 ^f^
G3	208	7	3.37 ^e^	200	6	3.00 ^de^
G4	195	9	4.62 ^b^	185	6	3.24 ^cd^
S1	235	2	0.85 ^h^	225	1	0.44 ^h^
S2	218	5	2.29 ^g^	208	5	2.40 ^f^
S3	203	5	2.46 ^g^	193	4	2.07 ^g^
S4	183	6	3.28 ^ef^	173	6	3.47 ^bc^
G1S1	230	7	3.04 ^f^	220	6	2.73 ^e^
G2S2	210	8	3.81^d^	200	6	3.00 ^de^
G3S3	192	8	4.17 ^c^	182	7	3.85 ^a^
G4S4	178	10	5.62 ^a^	168	6	3.57 ^b^

Numbers with the same superscripted letter are not different at the 5% significance level.

**Table 2 plants-11-01322-t002:** Frequency and spectrum of chlorophyll mutants induced in different doses of γ rays and SA in cowpea varieties Gomati V-89 and Pusa-578.

Doses	N	Var. Gomati VU-89							F (%)	k
		*Albina*	*Chlorina*	*Xantha*	*Tigrina*	*Viridis*	*Xanthviridis*	CMS		
C	3255	-	-	-	-	-	-	-	0.00 ^e^	-
G1	3205	12	8	5	1	3	-	29	0.90 ^bc^	-
G2	3146	13	7	5	-	4	-	29	0.92 ^bc^	-
G3	3102	10	8	4	2	4	5	33	1.06 ^b^	-
G4	3050	9	8	2	-	1	2	22	0.72 ^cd^	-
S1	3185	13	10	1	3	1	-	28	0.88 ^bc^	-
S2	3102	14	8	1	-	3	2	28	0.90 ^bc^	-
S3	3052	10	7	5	4	-	2	28	0.92 ^bc^	-
S4	2925	9	10	4	-	2	1	26	0.89 ^bc^	-
G1S1	3025	11	-	-	1	-	4	16	0.53 ^d^	0.30 ^d^
G2S2	2956	13	12	-	1	-	3	29	0.98 ^b^	0.54 ^c^
G3S3	2854	15	15	6	4	6	2	48	1.68 ^a^	0.85 ^b^
G4S4	2705	15	13	7	1	7	1	44	1.63 ^a^	1.02 ^a^
C	3147	**Var. Pusa-578**							0.00 ^g^	-
		-	-	-	-	-	-	-		
G1	3097	10	6	0	1	1	-	18	0.58 ^ef^	-
G2	3038	11	5	3	-	2	-	21	0.69 ^de^	-
G3	2994	9	5	3	2	2	2	23	0.77 ^cd^	-
G4	2942	7	7	0	-	2	1	17	0.58 ^ef^	-
S1	3077	11	10	1	4	3	-	29	0.94 ^bc^	-
S2	2994	13	10	2	-	2	2	29	0.97 ^b^	-
S3	2944	13	7	3	3	2	2	30	1.02 ^b^	-
S4	2817	11	9	2	-	2	1	25	0.89 ^bc^	-
G1S1	2917	10	-	-	1	-	3	14	0.48 ^f^	0.32 ^c^
G2S2	2848	14	9	-	2	-	2	27	0.95 ^bc^	0.57 ^b^
G3S3	2746	16	14	6	5	6	2	49	1.78 ^a^	0.99 ^a^
G4S4	2597	14	14	6	3	5	2	44	1.69 ^a^	1.15 ^a^

N, number of M_2_ seedlings; CMS, chlorophyll-mutated seedlings; F, frequency; k, interaction coefficient based on % M_2_ chlorophyll mutation frequency; Numbers with the same superscripted letter are not different at the 5% significance level.

**Table 3 plants-11-01322-t003:** Comparative frequency and spectrum of chlorophyll mutants (pooled data).

Mutagen	Var. Gomati VU-89						Frequency
	*Albina*	*Chlorina*	*Xantha*	*Tigrina*	*Viridis*	*Xanthviridis*	
γ rays	0.35 ^b^	0.25 ^a^	0.13 ^a^	0.02 ^b^	0.10 ^a^	0.06 ^a^	0.90
SA	0.38 ^ab^	0.29 ^a^	0.09 ^a^	0.06 ^a^	0.05 ^a^	0.04 ^a^	0.90
γ rays + SA	0.47 ^a^	0.35 ^a^	0.11 ^a^	0.06 ^a^	0.11 ^a^	0.09 ^a^	1.19
**Average Frequency**	0.40	0.30	0.11	0.04	0.09	0.06	1.00
γ rays	**Var. Pusa-578**						0.65
	0.31 ^a^	0.19 ^a^	0.05 ^a^	0.03 ^b^	0.06 ^a^	0.02 ^b^	
SA	0.41 ^a^	0.30 ^a^	0.07 ^a^	0.06 ^ab^	0.08 ^a^	0.04 ^b^	0.96
γ rays + SA	0.49 ^a^	0.33 ^a^	0.11 ^a^	0.10 ^a^	0.10 ^a^	0.08 ^a^	1.21
**Average Frequency**	0.40	0.27	0.07	0.06	0.09	0.04	0.94

γ, gamma; SA, sodium azide. Numbers with the same superscripted letter are not different at the 5% significance level.

**Table 4 plants-11-01322-t004:** Effectiveness and efficiency of γ rays, SA, and γ rays + SA treatments in cowpea varieties Gomati VU-89 and Pusa-578.

Mutagen Doses	Var. Gomati VU-89					Mp/I	Mp/S	Mp/Me
	Seedling Injury (I)	Pollen Sterility (S)	Meiotic Aberrations (Me)	Mutated Plant (Mp)	Effectiveness			
C	--	--	--	--	--	--	--	--
G1	10.91 ^g^	6.41 ^g^	3.19 ^f^	2 ^f^	0.02 ^e^	0.18 ^cde^	0.31 ^cd^	0.63 ^f^
G2	16.37 ^f^	7.69 ^fg^	5.14 ^de^	11 ^a^	0.06 ^e^	0.67 ^a^	1.43 ^a^	2.14 ^a^
G3	18.88 ^ef^	11.54 ^de^	5.47 ^de^	7 ^cde^	0.02 ^e^	0.37 ^bc^	0.61 ^bc^	1.28 ^bc^
G4	25.61 ^b^	15.90 ^c^	8.51 ^ab^	9 ^abc^	0.02 ^e^	0.35 ^bc^	0.57 ^bcd^	1.06 ^bcde^
S1	17.40 ^ef^	5.90 ^g^	1.63 ^fg^	2 ^f^	33.33 ^b^	0.11 ^de^	0.34 ^cd^	1.23 ^bcd^
S2	19.64 ^def^	9.74 ^ef^	3.50 ^ef^	5 ^e^	41.66 ^a^	0.25 ^bcd^	0.51 ^bcd^	1.43 ^b^
S3	22.31 ^cd^	15.38 ^c^	6.02 ^cd^	5 ^e^	27.77 ^c^	0.22 ^bcd^	0.33 ^cd^	0.83 ^def^
S4	29.13 ^a^	20.00 ^b^	7.63 ^bc^	6 ^de^	25.00 ^d^	0.21 ^bcd^	0.30 ^d^	0.79 ^ef^
G1S1	16.89 ^ef^	5.13 ^g^	5.44 ^de^	7 ^cde^	1.16 ^e^	0.41 ^b^	1.37 ^a^	1.29 ^bc^
G2S2	20.35 ^de^	10.77 ^de^	6.43 ^cd^	8 ^bcd^	0.33 ^e^	0.39 ^bc^	0.74 ^b^	1.25 ^bc^
G3S3	25.05 ^bc^	13.33 ^cd^	9.05 ^ab^	8 ^bcd^	0.14 ^e^	0.32 ^bcd^	0.60 ^bcd^	0.88 ^cdef^
G4S4	31.79 ^a^	22.82 ^a^	10.31 ^a^	10 ^ab^	0.10 ^e^	0.31 ^bcd^	0.44 ^cd^	0.97 ^cdef^
C	**Var. Pusa-578**					--	--	--
	--	--	--	--	--			
G1	12.28 ^e^	9.87 ^ef^	3.19 ^f^	1 ^c^	0.01 ^e^	0.08 ^cd^	0.10 ^fg^	0.31 ^de^
G2	17.17 ^cde^	11.14 ^ef^	5.77 ^bc^	6 ^ab^	0.03 ^e^	0.35 ^ab^	0.54 ^a^	1.04 ^abc^
G3	21.51 ^abc^	16.96 ^cd^	5.83 ^bc^	6 ^ab^	0.02 ^e^	0.28 ^ab^	0.35 ^bcd^	1.03 ^abc^
G4	23.47 ^ab^	21.01 ^b^	8.48 ^a^	6 ^ab^	0.02 ^e^	0.26 ^ab^	0.29 ^bcdef^	0.71 ^cd^
S1	11.99 ^e^	8.86 ^f^	1.05 ^g^	1 ^c^	16.66 ^d^	0.08 ^cd^	0.11 ^efg^	0.95 ^abc^
S2	14.84 ^de^	13.42 ^de^	3.56 ^ef^	5 ^ab^	41.66 ^a^	0.34 ^bc^	0.37 ^abcd^	1.41 ^a^
S3	19.18 ^bcd^	20.51 ^bc^	5.00 ^bcde^	4 ^b^	22.22 ^c^	0.21 ^bc^	0.20 ^def^	0.80 ^bc^
S4	20.12 ^bcd^	23.29 ^ab^	6.62 ^b^	6 ^ab^	25.00 ^b^	0.30 ^ab^	0.26 ^cdef^	0.91 ^abc^
G1S1	13.11 ^e^	10.89 ^ef^	4.05 ^def^	5 ^ab^	1.00 ^e^	0.38 ^a^	0.46 ^ab^	1.23 ^abc^
G2S2	17.72 ^bcde^	13.67 ^de^	4.90 ^cde^	6 ^ab^	0.25 ^e^	0.34 ^ab^	0.44 ^abc^	1.22 ^abc^
G3S3	21.12 ^bc^	21.77 ^b^	5.63 ^bcd^	7 ^ab^	0.13 ^e^	0.33 ^ab^	0.32 ^bcd^	1.24 ^abc^
G4S4	26.88 ^a^	26.33 ^a^	8.43 ^a^	8 ^a^	0.06 ^e^	0.30 ^ab^	0.30 ^bcde^	0.95 ^abc^

Mp/I, efficiency based on seedling injury; Mp/S, efficiency based on pollen sterility; Mp/Me, efficiency based on meiotic abnormalities. Numbers with the same superscripted letter are not different at the 5% significance level.

**Table 5 plants-11-01322-t005:** Frequency and spectrum of morphological mutants in cowpea varieties Gomati VU-89 and Pusa-578.

Doses	N	Var. Gomati VU-89							% Mutated Plants	k
		Plant Height Mutants	Growth Habit Mutants	Leaf Mutants	Flower Mutants	Pod Mutants	Seed Mutants	Total Mutated Plants		
C	3050	-	-	-	-	-	-	-	-	-
G1	3000	8	-	6	2	9	4	29	0.97 ^e^	-
G2	2941	4	10	13	5	-	5	37	1.26 ^d^	-
G3	2897	-	16	9	7	-	6	38	1.31 ^cd^	-
G4	2845	8	8	-	2	6	3	27	0.95 ^e^	-
**Total**	**11,683**	**20**	**34**	**28**	**16**	**15**	**18**	**131**	**4.49**	-
S1	2980	9	-	11	12	-	3	35	1.17 ^de^	-
S2	2897	5	-	6	5	11	6	33	1.14 ^de^	-
S3	2847	-	12	7	10	-	7	36	1.26 ^d^	-
S4	2720	10	7	13	8	7	4	49	1.80 ^b^	-
**Total**	**11,444**	**24**	**19**	**37**	**35**	**18**	**20**	**153**	**5.38**	-
G1S1	2820	6	-	-	15	-	9	30	1.06 ^de^	0.49 ^c^
G2S2	2751	6	-	19	13	12	10	60	2.18 ^a^	0.90 ^a^
G3S3	2649	-	16	9	-	-	4	29	1.09 ^de^	0.42 ^d^
G4S4	2500	-	17	7	-	14	0	38	1.52 ^c^	0.55 ^b^
**Total**	**10,720**	**12**	**33**	**35**	**28**	**26**	**23**	**157**	**5.86**	**2.36**
C	2960	**Var. Pusa-578**							-	-
		-	-	-	-	-	-	-		
G1	2910	10	-	8	2	-	9	29	1.00 ^f^	-
G2	2851	2	9	14	4	11	10	50	1.75 ^bc^	-
G3	2807	-	14	8	5	-	5	32	1.14 ^ef^	-
G4	2755	7	10	-	4	8	3	32	1.16 ^ef^	-
**Total**	**11,323**	**19**	**33**	**30**	**15**	**19**	**27**	**143**	**5.05**	**0.00**
S1	2890	10	-	11	12	-	4	37	1.28 ^ef^	-
S2	2807	7	13	6	5	9	8	48	1.71 ^bcd^	-
S3	2757	-	-	8	10	9	9	36	1.31 ^def^	-
S4	2630	11	8	10	10	-	7	46	1.75 ^bc^	-
**Total**	**11,084**	**28**	**21**	**35**	**37**	**18**	**28**	**167**	**6.05**	**0.00**
G1S1	2730	8	-	-	14	-	10	32	1.17 ^ef^	0.51 ^b^
G2S2	2661	8	-	17	15	13	11	64	2.41 ^a^	0.70 ^a^
G3S3	2559	3	15	11	-	2	5	36	1.41 ^cdef^	0.57 ^b^
G4S4	2410	-	14	9	-	15	0	38	1.58 ^bcde^	0.58 ^b^
**Total**	**10,360**	**19**	**29**	**37**	**29**	**30**	**26**	**170**	**6.56**	**2.42**

N, number of plants; k, coefficient of interaction. Numbers with the same superscripted letter are not different at the 5% significance level.

**Table 6 plants-11-01322-t006:** Frequency and spectrum of morphological mutants in the M_2_ generation of cowpea var. Gomati VU-89.

Characters	Morphological Mutants	Var. Gomati VU-89								Grand Total	
		γ Rays		SA		γ Rays + SA		Total			
		N	F%	N	F%	N	F%	N	F%	N	F%
**Plant height**	Tall	13	0.11	15	0.13	9	0.08	37	0.11	**56**	**0.17 ^ab^**
	Dwarf	5	0.04	8	0.07	2	0.02	15	0.04		
	Semi-Dwarf	2	0.02	1	0.01	1	0.01	4	0.01		
**Growth habit**	Bushy	8	0.07	7	0.06	12	0.11	27	0.08	**63**	**0.19 ^a^**
	Prostrate	1	0.01	2	0.02	5	0.05	8	0.02		
	Semi-Dwarf Spreading	3	0.03	4	0.03	3	0.03	10	0.03		
	One-Sided Branching	7	0.06	1	0.01	1	0.01	9	0.03		
	Axillary Branching	3	0.03	2	0.02	4	0.04	9	0.03		
**Leaf**	Broad Leaf	8	0.07	7	0.06	6	0.06	21	0.06	**59**	**0.17 ^ab^**
	Narrow Leaf	5	0.04	5	0.04	7	0.07	17	0.05		
	Altered Leaf Architecture	2	0.02	2	0.02	5	0.05	9	0.03		
	Elongated Rachis	4	0.03	6	0.05	2	0.02	12	0.04		
**Flower**	Multi Flowering	4	0.04	3	0.03	3	0.03	10	0.03	**66**	**0.19 ^a^**
	Flower Color	2	0.02	5	0.04	4	0.04	11	0.03		
	Open Flower	7	0.06	7	0.06	6	0.06	20	0.06		
	Non-Flowering	1	0.01	1	0.01	2	0.02	4	0.01		
	Late Flowering	6	0.05	6	0.05	2	0.02	14	0.04		
	Early Maturity	4	0.03	2	0.02	1	0.01	7	0.02		
**Pod**	Narrow Pod	7	0.06	8	0.07	10	0.09	25	0.07	**45**	**0.13 ^b^**
	Broad Pod	5	0.04	6	0.05	9	0.08	20	0.06		
**Seed**	Coat Color	7	0.06	8	0.07	7	0.07	22	0.06	**75**	**0.22 ^a^**
	Coat Pattern	6	0.05	9	0.08	8	0.07	23	0.07		
	Shape and Surface	10	0.09	11	0.10	9	0.08	30	0.09		
**Grand Total**		**120**	**1.03**	**126**	**1.10**	**118**	**1.10**	**364**	**1.08**		

N, number of M_2_ plants; γ, gamma; SA, sodium azide; F, frequency. Numbers with the same superscripted letter are not different at the 5% significance level.

**Table 7 plants-11-01322-t007:** Frequency and spectrum of morphological mutants in the M_2_ generation of cowpea var. Pusa-578.

Characters	Morphological Mutants	Var. Pusa-578								Grand Total	
		γ Rays		SA		γ Rays + SA		Total			
		N	F%	N	F%	N	F%	N	F%	N	F%
**Plant height**	Tall	10	0.09	11	0.10	8	0.08	29	0.09	**44**	**0.13 ^a^**
	Dwarf	4	0.04	7	0.06	1	0.01	12	0.04		
	Semi-Dwarf	1	0.01	1	0.01	1	0.01	3	0.01		
**Growth habit**	Bushy	7	0.06	6	0.05	11	0.11	24	0.07	**57**	**0.17 ^a^**
	Prostrate	4	0.04	1	0.01	6	0.06	11	0.03		
	Semi-Dwarf Spreading	2	0.02	3	0.03	2	0.02	7	0.02		
	One-Sided Branching	5	0.04	2	0.02	1	0.01	8	0.02		
	Axillary Branching	2	0.02	3	0.03	2	0.02	7	0.02		
**Leaf**	Broad Leaf	7	0.06	6	0.05	5	0.05	18	0.05	**50**	**0.15 ^a^**
	Narrow Leaf	4	0.04	4	0.04	5	0.05	13	0.04		
	Altered Leaf Architecture	1	0.01	3	0.03	4	0.04	8	0.02		
	Elongated Rachis	5	0.04	5	0.05	1	0.01	11	0.03		
**Flower**	Multi Flowering	3	0.03	2	0.02	4	0.04	9	0.03	**64**	**0.20 ^a^**
	Flower Color	3	0.03	4	0.04	5	0.05	12	0.04		
	Open Flower	6	0.05	8	0.07	5	0.05	19	0.06		
	Non-Flowering	2	0.02	2	0.02	1	0.01	5	0.02		
	Late Flowering	5	0.04	5	0.05	1	0.01	11	0.03		
	Early Maturity	3	0.03	4	0.04	1	0.01	8	0.02		
**Pod**	Narrow Pod	5	0.04	7	0.06	9	0.09	21	0.06	**45**	**0.14 ^a^**
	Broad Pod	4	0.04	5	0.05	8	0.08	17	0.05		
**Seed**	Coat Color	6	0.05	7	0.06	6	0.06	19	0.06	**75**	**0.23 ^a^**
	Coat Pattern	5	0.04	8	0.07	7	0.07	20	0.06		
	Shape and Surface	9	0.08	10	0.09	8	0.08	27	0.08		
**Grand Total**		**103**	**0.91**	**114**	**1.03**	**102**	**0.98**	**319**	**0.97**		

N, number of M_2_ plants; γ, gamma; SA, sodium azide; F, frequency. Numbers with the same superscripted letter are not different at the 5% significance level.

## Data Availability

The data is contained within this article and supplementary file.

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
