# Peer review of "Comparative Mutagenic Effectiveness and Efficiency of Gamma Rays and Sodium Azide in Inducing Chlorophyll and Morphological Mutants of Cowpea"

_plants, 2022, doi:10.3390/plants11101322_

Round 1
Reviewer 1 Report
Dear authors, I accept the manuscript for publication after minor revision and attach a file with my suggestions.

Author Response
Esteemed Editors and Reviewers:
We are highly grateful for your painstaking comments. We have addressed the comments and incorporated suggestions to the best of our knowledge. Your comments and suggestions have increased the worth of our manuscript. Please find below our point-by-point response
The manuscript “Comparative mutagenic effectiveness and efficiency of gamma rays and sodium azide in inducing chlorophyll and morphological mutants of cowpea” by Aamir Raina, Rafiul Amin Laskar, Mohammad Rafiq Wani, Samiullah Khan, Parvaiz Ahmad makes a contribution to the scientific knowledge enrichment in mutagenesis area. Different individual and combined mutagens are evaluated for inducing various mutants of Vigna unguiculata (L.) which are detailed described. This is a large-scale study that meets the requirements of mutation breeding. The manuscript is clear, well-structured, and relevant for the field and can be accepted for publication after minor changes.
Response: Thank you for your encouraging words.
Title - explicit and matches the content.
Abstract - enough informative.
Keywords:
There is repetition of some words from the title. I suggest: Vigna unguiculata (L.), mutation frequency; seed germination, pollen sterility, coefficient of interaction, M1/M2 generations
Response: We have incorporated keywords as suggested by the Respected Reviewer.
Introduction
The necessity of this study with cowpea is clarified and different studies on the base of mutagenesis conducted by other research groups are summarized.
Response: Thank you for your nice and encouraging words.
Results
This part is well organized, illustrated and provides explains of the results of the experiment. Nevertheless I would like to share my notes to improve the article:
Line 94-95 In M2 generation of var. Gomati VU-89 there are only non-significant decreases in seed germination of all applied mutagen doses (Fig. 1) which must be mentioned.
Response: We have added the required details (Line 92-93; 99-101)
Also must be noted in var. Gomati VU-89 only reduction of pollen fertility using G4, S4 and G4S4 is significant (Fig. 3) while in var. Pusa-578 there are significant reductions when six mutagen are applied - G3, G4, S3, S4, G3S3, G4S4 (Fig. 4).
Response: We have added the required details (Line 111-117)
There are some wrong calculated values about coefficient of interaction of mutagens k (%) which are different from cited formula. For example: Line 204 Table 2 for var. Pusa-578 G3S3 k is 0.99 %, not 1.02%. In Table 5 all values for k are not correct for both varieties.
Response: Thank you for this insightful comment. We have now corrected the values in both Table 2 and Table 5.
S 1Table – in var. Pusa-578 there is mistake, must be selection from, not form.
Response: We have corrected it.
S2 Table – for C treated seeds put asterisk and in legend must be noted that 300 seeds also are used for control.
Response: We put asterisk and mentioned in the legend that that 300 seeds were also used for control.
Line 376 – More explanations about control plants, leaves colour, size of internodes.
Response: Thank you for your comment. However, we would like to mention that we have added all the details of control plants such as their plant height, growth habit, leaf, flower, seed and pod attributes in the respective sections of mutants.
Line 379 – The differences between tall mutants and control plants in height are very small (only 2.5 cm)
Response: We calculated the values in the field and we were also surprised to observe the small difference between tall mutants and control.
Discussion
The authors well discuss the obtained results and combine with their previous and other studies.
Response: Thank you for your encouraging words.
Conclusions - according to the obtained results
Response: Thank you for your encouraging words.
Materials and methods
In this part all materials and methods are detailed described. The experiments are well designed and statistical analysis is adequate.
Response: Thank you for your encouraging words.
References
The cited references are mostly within the last 15 years. The list is comparative up to date, properly formatted.
Response: Thank you for your encouraging words.

Reviewer 2 Report
In the MS "Comparative mutagenic effectiveness and efficiency of gamma rays and sodium azide in inducing chlorophyll and morphological mutants of cowpea" the authors tested the effect of different mutation doses in cowpea, identifying M2 mutants with desirable characteristics for future breeding programs.
The manuscript is well structured and the experimental design is appropriate.
However, the MS needs revision of english style and typos.
Please, carefully go though the attached file evaluating modifications in the highlighted parts.

Author Response
Esteemed Editors and Reviewers:
We are highly grateful for your painstaking comments. We have addressed the comments and incorporated suggestions to the best of our knowledge. Your comments and suggestions have increased the worth of our manuscript. Please find below our point-by-point response
A total of 57620 M2 seeds were generated from the M1 generation of two varieties—Gomati VU-89 and Pusa-578 in which 47650 (82.69%) seeds germinated, of which 38749 (81.32%) plants survived were screened for chlorophyll and morphological mutations.
Response: We have reframed the sentence.
In the present era of climate change led extreme weather events such as, inconsistent rainfalls, depleting arable land, exhausting water resources all pose a significant risk to the agricultural productivity. Besides climate change a rapidly growing population that is expected to rise to 9.6 billion by 2050 imposes huge pressure on the agriculture and its allied sectors.
Response: We have corrected the sentence.
Therefore, pulses are ideal crops to meet the food 46 and nutritional demands of rapidly growing population and sustainable development goal 2 that ensure zero hunger for all people
Response: We have reworded the sentence.
Among the ten primary pulse crops recognized by Food and Agriculture Organisation (FAO), cowpea [Vigna unguiculata (L.) Walp.] is an important member and man's primaeval pulse crops based on its use, nutritional value and other desired qualities [4]
Response: We have reframed the sentence.
World-56 wide 14.4 million hectares of land is devoted to cowpea cultivation, producing about 8.9 million tonnes
Response: We have reworded the sentence.
The γ rays and SA were chosen as mutagens based on their high effectiveness and efficiency in inducing the mutations.
Response: We have corrected the sentence.
The optimum mutagen doses have successfully developed and 76 officially released several hundred improved mutant varieties [13]. Fifteen mutant varieties of cowpea with improved agronomic traits have been developed.
Response: We have added the reference.
It is evident from the literature, a variable ranges of γ rays have been used to enhance yield without a shred of conclusive evidence on optimum dose of γ rays that could be employed anywhere to improve the yielding potential of cowpea.
Response: We have reframed the sentence.
The present study of induced mutagenesis was undertaken under field conditions from April 2014 to October 2017.
Response: We have shifted the selected portion to the materials and method section.
The number of M1 plants progenies, segregating progenies, and per cent of mutated plants are given in table 1.
Response: We have reframed the sentence.
In both the varieties germination inhibition percent showed a dose-dependent increase with the increase in mutagenic concentration.
Response: We have corrected the sentence.
In contrast micro mutations defined as quantitatively inherited, and phenotypically invisible alterations included plant height, growth habit, and pod mutations.
Response: We have reframed the sentence.
